# Egg White Ovotransferrin Attenuates RANKL-Induced Osteoclastogenesis and Bone Resorption

**DOI:** 10.3390/nu11092254

**Published:** 2019-09-19

**Authors:** Nan Shang, Jianping Wu

**Affiliations:** Department of Agricultural, Food and Nutritional Science, University of Alberta, Edmonton, AB T6G 2P5, Canada; nshang@ualberta.ca

**Keywords:** osteoclasts, osteoclastogenesis, NF-κB pathway, MAPK pathway, apoptosis

## Abstract

Ovotransferrin, a member of the transferrin family, is the second main protein found in egg white. Ovotransferrin was reported to have antimicrobial, antioxidant, and immunomodulating activities. The aim of this work was to characterize the cellular and molecular functions of egg white ovotransferrin on osteoclasts differentiation and function. Osteoclasts were prepared from mouse macrophage RAW 264.7 cells stimulated with receptor activator of nuclear factor κB ligand (RANKL). Ovotransferrin inhibited osteoclasts differentiation and the calcium–phosphate resorptive ability via the suppression of RANKL-induced nuclear factor κ-light chain-enhancer of activated B cells (NF-κB) and mitogen-activated protein kinase (MAPK) signaling pathways. Ovotransferrin induced apoptosis of matured osteoclasts, accompanied by increased expression of Bcl-2-like protein 11 (*Bim*) and Bcl-2-assoicated death promoter (*Bad*), but decreased expression of B-cell lymphoma 2 (*Bcl-2*) and B-cell lymphoma-extra-large (*Bcl-xl*). We established a novel role of egg white ovotransferrin as an inhibitor of osteoclastogenesis, which may be used for the prevention of osteoporosis.

## 1. Introduction

Bone is a dynamic organ that undergoes continuous remodeling to change its mass and form [1]. When healthy, bone remodeling maintains a delicate balance, in which the amounts of total bone resorption regulated by osteoclasts and total bone formation regulated by osteoblasts remain the same [2]. However, over-activated osteoclasts activity can lead to osteolytic bone conditions, resulting in several bone diseases such as osteoporosis [2]. The decline in bone mass and the deterioration of its architectural integrity are considered the consequences of the massive recruitment of osteoclasts and excessive osteoclastic activity [2,3]. Thus, the inhibition of osteoclastic activity and the resultant bone resorption have a profound effect on regulating abnormal remodeling processes and is therapeutically important for osteoporosis prevention.

Osteoclasts, which function to resorb bone, are giant and multinucleated cells derived from monocytic and macrophage lineage [4]. To resorb bone, the generation and differentiation of osteoclasts from their precursors are crucial. The essential step in osteoclasts generation is the binding of the receptor-activator of nuclear factor κ-light chain-enhancer of activated B cells (NF-κB) ligand (RANKL) to its receptor, receptor-activator of NF-κB (RANK), in osteoclast precursors. Once RANKL binds to its receptor RANK, the downstream osteoclastogenesis signaling pathways, such as such as NF-κB, proto-oncogene tyrosine-protein kinase Src (c-Src), and mitogen-activated protein kinase (MAPK) pathways, including p38 MAPK, c-jun-N-terminal kinase (JNK), and extracellular signal-regulated kinase (ERK), are activated by the trimerization of tumor necrosis factor (TNF) receptor-associated factor 6 (TRAF6) [5,6]. Activation of NF-κB and MAPK pathways consequently leads to increased expression of transcription factors, such as proto-oncogene c-Fos (c-Fos) and the nuclear factor of activated T-cells, cytoplasmic 1 (NFATc1), which are vital for the differentiation of osteoclast precursors [5]. Hence, targeting NF-κB and MAPK/activator protein 1 (AP-1) signaling pathways to inhibit osteoclastogenesis has been considered a promising strategy for osteoporosis treatment.

In addition to impairing osteoclasts formation, the induction of cell apoptosis of mature osteoclasts is a well-accepted method for osteoporosis treatment. Genetically modified animal models have identified the important relationships between osteoclast apoptosis and osteolytic disorders [7]. In general, decreased osteoclasts apoptosis results in increased bone loss, such as in the ovariectomy-induced estrogen deficiency model [8,9]. Reduced osteoclasts apoptosis in inflamed joints partly contributes to the inefficiency of bisphosphonate therapy in osteoporosis treatment in patients with rheumatoid arthritis [10]. Apoptosis is generally activated by two different pathways: the intrinsic pathway, which is activated through mitochondria, and the extrinsic pathway, which is recognized by ligand activation via death receptors [7,11,12]. Although the extrinsic-pathway-regulated osteoclast apoptosis shows some difference compared with other cell types [10,13,14], osteoclasts contain the general apoptosis machinery of the intrinsic pathway [7]. Caspases are one of the main adhesion molecules involved in cell apoptosis [15]; their activation and release are largely modulated by B-cell lymphoma 2 (Bcl-2) family members through the mitochondria [16]. The Bcl-2 family consists of a group of proteins that either promote cell apoptosis (apoptotic members), such as Bad, Bax, and Bid, or inhibit apoptosis (anti-apoptotic members), such as Bcl-2 and B-cell lymphoma-extra-large (Bcl-xl) [11]. Among Bcl-2 family members, Bcl-xl is one of the proteins that has been most studied in osteoclasts. Macrophage colony-stimulating factor (M-CSF), RANKL, and TNF, identified as osteoclastogenesis triggers, can significantly promote Bcl-xl expression in completely differentiated osteoclasts [7,17]. Over-expression of Bcl-xl in osteoclasts in vitro protects against bisphosphonate-induced cell death, which leads to a prolonged osteoclasts life span and therefore interferes with osteoporosis treatment [18]. Increased Bcl-2 expression and ratio of Bcl-2 to Bax have been found in Pagetic osteoclasts, which significantly contribute to osteoclast formation and maturation. More multinucleate osteoclasts have been observed in Pagetic osteoclasts rather than normal status, which can lead to increased osteoclastic bone resorption. Abnormally high expression of Bcl-2 protein also postponed the apoptosis of matured osteoclasts in Pagetic patients [19,20].

To date, most FDA (Food and Drug Administration)-approved medications against osteoporosis are anti-resorption drugs, such as bisphosphonates. These anti-resorptive agents prevent bone loss by inhibiting osteoclasts formation and/or inducing apoptosis in mature osteoclasts, but are associated with severe side effects such as renal system impairment and induction of osteonecrosis in the jaw [21]. Food-derived natural products are gaining the momentum in the development of alternative treatment options against osteoporosis. Ovotransferrin is an iron-binding glycoprotein, which constitutes about 12% of the total egg white protein. Egg ovotransferrin has many biological activities, such as antimicrobial, antioxidant, and immunomodulating activities [22]. Our previous study reported that ovotransferrin is able to stimulate osteogenic activity via regulation of the RANKL/OPG ratio [23], suggesting its potential ability to mediate osteoclastogenesis. In the present study, we hypothesized that ovotransferrin could help prevent RANKL-induced osteoclastogenesis and induce apoptosis in mature osteoclasts. Hence, we tested the activity of ovotransferrin in a cell model of osteoclastogenesis and identified the underlying molecular mechanisms. 

## 2. Materials and Methods

### 2.1. Reagents and Antibodies

Recombinant mouse soluble RANKL (Gene ID: 21943) was obtained from R&D Systems (Catalog number: 462-TEC-010, Oakville, ON, Canada). Dulbecco modified Eagles medium (DMEM), fetal bovine serum (FBS), penicillin-streptomycin (10,000 U/mL), and TRIzol Reagent were purchased from Thermo Fisher Scientific (Burlington, ON, Canada). Ovotransferrin (conalbumin from chicken egg white) with purity ≥98% was purchased from Sigma-Aldrich (Catalog number: C0755, St. Louis, MO, USA). The tartrate-resistant acid phosphatase (TRAP)-staining kit was obtained from Sigma-Aldrich (Catalog number: 387A-1KT, St. Louis, MO, USA). The annexin V-FITC Apoptosis Staining/Detection Kit was purchased from Abcam (Catalog number: ab14085, Toronto, ON, Canada). The NF-κB pathway sampler kit (Catalog number: 9936T), MAPK family antibody sampler kit (Catalog number: 9926T), and phosphor-MAPK family antibody sampler kit (Catalog number: 9910T) were purchased from Cell Signaling Technology (Whitby, ON, Canada). Recombinant anti-TRAF6 antibody (Catalog number: ab33915), anti-c-Fos antibody (Catalog number: ab190289), and anti-α tubulin antibody (Catalog number: ab7291) were purchased from Abcam (Cambridge, MA, USA). NFATc1 antibody (Catalog number: 7A6) and cathepsin K antibody (Catalog number: E-7) were purchased from Santa Cruz Biotechnology (Dallas, TX, USA).

### 2.2. Cell Culture

The murine monocyte cell line RAW 264.7 (ATCC TIB-71) was obtained from ATCC (Manassas, VA, USA). RAW 264.7 cells were cultured in DMEM supplemented with 10% FBS and 1% pen-strep at 37 °C with 5% CO_2_ atmosphere. All experiments were established between passage numbers from 18 to 25.

### 2.3. Osteoclasts Origination and TRAP Staining

Osteoclast cells were induced from murine monocyte cell line RAW 264.7. RAW 264.7 cells were seeded in a cell culture plate at a density of 1 × 10^4^–5 × 10^4^ cells/well and incubated for 6 h until the cells were attached to the plate surface. After 6 h, DMEM containing 100 ng/mL RANKL were used to replace the old culture medium. Osteoclasts were successfully induced by incubating with RANKL for 4 days. RAW 264.7 cells were treated with different concentrations of ovotransferrin (1 to 1000 μg/mL) for 4 days to examine the effects of ovotransferrin on osteoclastogenesis. To confirm osteoclast generation and count the number of multinucleated osteoclast-like cells, cells were stained by the enzyme tartrate-resistant acid phosphatase (TRAP) using the TRAP-staining kit according to the manufacturer’s instructions. Briefly, cells were fixed in 4% paraformaldehyde for 1 h at 4 °C and then stained with TRAP staining solution (0.1 mg/mL naphthol AS-MX phosphate and 0.3 mg/mL fast red violet LB staining). The cells were observed with a light microscope under a 10× lens (Olympus IX83, Richmond Hill, ON, Canada) and the images were captured using Metamorphy (Olympus, Richmond Hill, ON, Canada). Any TRAP-positive multinucleated cells containing three or more nuclei were identified as mature osteoclasts. 

### 2.4. Cell Apoptosis Assay

Osteoclasts were generated by stimulating RAW 264.7 cells by RANKL (100 ng/mL) for 4–6 days as described in Section 2.3 and identified by microscopy. Then, cells were treated with 1–1000 μg/mL ovotransferrin for 12 h. After treatment, cells were trypsinized and collected by centrifugation at 500× *g* for 5 min. Apoptosis was detected using the apoptosis detection kit following the manufacturer’s instructions. Briefly, we re-suspended cells in 500 μL binding buffer and added 5 μL of Annexin V-Fluorescein isothiocyanate (FITC) and 5 μL of propidium iodide for 5 min at room temperature with protection against light. The Annexin V-FITC/PI binding was analyzed by flow cytometry (FACSCanto II, BD Biosciences, Mississauga, ON, Canada).

### 2.5. Resorption Assay

RAW 264.7 cells were incubated in a calcium phosphate (CaP)-coated 48-well plate (Cosmo Bio Co., Ltd., Tokyo, Japan) with 100 ng/mL RANKL to generate multinucleated osteoclasts, and treated with 1–1000 μg/mL ovotransferrin for 6 days. We added 150 μL new medium into each well every 2 days without removing old culture medium. The generated multinucleated osteoclasts were identified with microscopy every day. We used 100 μL culture medium to identify the resorption by measuring the fluorescence intensity every 2 days. After 6 days, all cells were eliminated using 5% hypochlorite for 5 min. Then, the plate was washed with Milli-Q water (Milli-Q water purification system, Millipore Ltd, Etobicoke, ON, Canada) and dried in air. The images of each well were captured using a microscope and the pit areas were calculated using Image J (LOCI, University of Wisconsin-Madison, Madison, WI, USA).

### 2.6. Western Blot Analysis

The cells were seeded on 48-well tissue culture plates at a concentration of 1 × 10^4^ cells/well, and cultured in DMEM with 10% FBS for 6 h to allow the cells to attach to the surface. Then, RAW 264.7 cells were pretreated with ovotransferrin (1–1000 μg/mL) for 2 h prior to stimulation with RANKL (100 ng/mL) and incubated together for 12 h. After removing the culture medium, cells were lysed in boiling hot Laemmle’s buffer with 50 μM dithiothreitol (DTT) and 0.2% Triton-X-100 and used for Western blot analysis as described previously [23]. These cell lysates were run in SDS (sodium dodecyl sulfate)-PAGE (polyacrylamide gel electrophoresis), blotted to nitrocellulose membranes, and immunoblotted with different antibodies described in Section 2.1. The dilution factor was 1:1000 for most of the antibodies, except for phosphor-p44/42 MAPK (1:2000), Cathepsin K (1:2000), TRAF6 (1:200), NFATc1 (1:200) and α-Tubulin (1:100,000). All antibodies were diluted into phosphate buffer saline (PBS) containing 5% tween 20. The α-tubulin was used as the internal reference. The protein bands were detected using a Licor Odyssey BioImager and quantified by densitometry using the corresponding software (Licor Biosciences, Lincoln, NB, USA). The results are expressed as percentage of the corresponding untreated control.

### 2.7. RNA Extraction and Quantitative PCR

Total RNA was extracted from the cells using TRIzol^®^ reagent (Thermo Fisher Scientific, Burlington, ON, Canada) following the manufacturer’s instructions, and cDNA was synthesized using SuperScript II Reverse Transcriptase (Invitrogen, Burlington, ON, Canada) according to the manufacturer’s instructions. Real-time PCR (RT-PCR) was performed using Fast SYBR Green PCR Master Mix (Applied Biosystems, Burlington, ON, Canada) and an ABI 7300 Sequencing Detection System (Applied Biosystems, Burlington, ON, Canada). The PCR amplification conditions were: initial denaturation at 95 °C for 10 min, followed by 35 cycles of 10 s at 95 °C, 15 s at 60 °C, and 10 s at 72 °C. The real-time PCR primers used in this study are listed in Table 1.

### 2.8. Statistical Analysis

All data are presented as mean ± standard error of the mean (SEM) and all results were obtained with 3 to 6 independent experiments. Data were analyzed by one-way analysis of variance (ANOVA) with Dunnett’s post-hoc test with comparisons to every other group. PRISM 6 statistical software (GraphPad Software, San Diego, CA, USA) was used for the analyses. *p* < 0.05 was considered significant.

## 3. Results

### 3.1. Ovotransferrin Inhibits Osteoclast Formation from RAW 264.7

To examine the effect of ovotransferrin on osteoclast formation from mouse osteoclast precursors, RAW 264.7 was incubated in osteoclastogenic medium (DMEM containing 100 ng/mL RANKL) with various concentrations of ovotransferrin (1–1000 μg/mL) for six days. RANKL induced osteoclast formation, whereas ovotransferrin addition significantly inhibited RANKL-induced formation of osteoclasts (Figure 1A). The total number of TRAP-positive (TRAP^+^) osteoclasts confirmed the inhibition effect of ovotransferrin on RANKL-induced osteoclast differentiation, although only high doses of ovotransferrin (100 and 1000 μg/mL) produced significant effects (Figure 1B). Cytotoxicity of ovotransferrin was evaluated using the Alarma Blue assay, which showed that this inhibition effect is not related to cytotoxicity (data not shown).

### 3.2. Ovotransferrin Inhibits RANKL-Stimulated Activation of NF-κB and MAPK Pathway

RANKL-stimulated osteoclast differentiation includes the activation of NF-κB and MAPK pathways via triggering the phosphorylation of several key proteins. We discovered that RANKL significantly boosted the phosphorylation of IKKα/β (Ser 176/178), IκBα (Ser 32), and NF-κB p65 (Ser 536), where ovotransferrin pre-treatment suppressed their phosphorylation (Figure 2). During osteoclast differentiation, RANKL also activated MAPK signaling as a synergistic action. Therefore, we further explored the effect ovotransferrin on MAPK activation under RANKL stimulation in addition to NF-κB. At high concentrations (100 and 1000 μg/mL), ovotransferrin significantly inhibited RANKL-induced phosphorylation of MAPKs (Figure 3). These data indicate that ovotransferrin inhibits osteoclast formation and differentiation by attenuating the activation of NF-κB and MAPK pathways.

### 3.3. Ovotransferrin Inhibits RANKL-Induced Expression of Proteins Involved in Osteoclastogenesis

We further evaluated the expression of the proteins involved in osteoclastogenesis including TRAF6, c-Fos, NFATc1, and cathepsin K. Western blotting indicated that RANKL significantly increased the expression of TRAF6, NFATc1, and cathepsin K. Pre-treatment with ovotransferrin significantly attenuated this increased expression of osteoclast-specific proteins (Figure 4). Although the RANKL stimulation did not increase the expression of c-Fos significantly, pre-treatment with ovotransferrin still showed significant decrease in c-Fos expression (Figure 4C). These data suggest that ovotransferrin inhibits osteoclast differentiation by inhibiting several osteoclastogenesis-related proteins.

### 3.4. Ovotransferrin Inhibits Resorptive Activity of Matured Osteoclasts

Since ovotransferrin prevented osteoclastogenesis by inhibiting RANKL-increased expression of osteoclastic markers, we further studied the effect of ovotransferrin on osteoclastic resorption. RAW 264.7 cultured in a fluoresceinated calcium phosphate (CaP)-coated plate was stimulated with 100 ng/mL RANKL and treated with different concentrations of ovotransferrin (1–1000 μg/mL) for six days to generate multinucleated osteoclasts. RANKL stimulation increased the fluorescent intensity of the culture medium with increasing time (Figure 5A), indicating the generation of osteoclasts and osteoclastic resorption of CaP; whereas ovotransferrin significantly reduced the osteoclastic resorption (Figure 5A,B). After six days, the total resorption pit areas were measured and the formation of resorbed pits were substantially reduced in the presence of ovotransferrin (Figure 5C,D). These data revealed that ovotransferrin could inhibit the bone-resorbing activity of mature osteoclasts in vitro in addition to preventing osteoclast differentiation.

### 3.5. Ovotransferrin Stimulates the Apoptosis of Osteoclasts

We further studied ovotransferrin-induced cell apoptosis of osteoclasts using flow cytometry. Mature multinucleated osteoclasts were identified using a microscope and then treated with different concentrations of ovotransferrin (1–1000 μg/mL). Flow cytometry revealed that treatment with high concentrations of ovotransferrin (100 and 1000 μg/mL) could induce cell apoptosis in mature osteoclasts (Figure 6A,B). Next, we further assessed the mRNA expression of *Bcl-2* family, which is one of the major regulators of cell apoptosis. RT-PCR analysis indicated that ovotransferrin treatment significantly amplified the gene expression of *Bim* and *Bad*, but decreased *Bcl-2* and *Bcl-xl* expression (Figure 6C). These findings suggest that ovotransferrin regulates the *Bcl-2* family, and may further act on osteoclasts mitochondria and induce apoptosis.

## 4. Discussion

Nutrients are important contributors to bone architecture and function; for example, vitamin D and calcium are well-studied bone management agents due to their beneficial effects on bone matrix development [24]. High-quality dietary protein consumption has been suggested to play a positive role in bone health [25]; a proper increase in food protein intake was reported to contribute to higher bone mineral density and stronger bone quality [25,26]. Food-derived bioactive proteins have been identified as having the potential to be used in therapeutic approaches in bone health research. For instance, soy proteins have been suggested to improve bone quality by inhibiting osteoclast activity [27,28]. Lactoferrin, derived from milk, was also reported to promote bone formation by stimulating osteoblastic activity [29] while preventing bone resorption by inhibiting osteoclast differentiation and resorptive activity [30]. In our previous study, we reported for the first time that the egg white protein ovotransferrin could directly promote osteoblasts proliferation and differentiation as an osteogenic agent [23]. 

Bone health largely relies on the remodeling process, which is mediated by bone formation cells (osteoblasts) and the bone resorption cells (osteoclasts) [1]. Abnormal osteoclastic activity leads to impairment of bone remodeling; for example, excessive bone resorption caused by over-activation of osteoclasts is a characteristic of several bone lytic disorders, such as osteoporosis. Thus, osteoclasts remain one of the crucial targets for developing osteoporosis therapies. Therefore, we aimed to illustrate the cellular and molecular effects of egg white ovotransferrin on osteoclastic activity and the associated bone resorption. We demonstrated that ovotransferrin exerts an anti-osteoclastogenesis effect via suppressing RANKL activation of NF-κB and MAPK signaling pathways. We also found that ovotransferrin could induce cell apoptosis in mature multinuclear osteoclasts by regulating the expression Bcl-2 family members (Figure 7).

Osteoclastogenesis is a complicated process that is primarily triggered by the binding of RANKL and RANK. The binding of RANKL to its receptor RANK is the initial and essential step in osteoclast generation, differentiation, and maturation. It initiates the recruitment and activation of TRAF6, and consequently activates downstream signaling, such as NF-κB and MAPKs (ERK, JNK, and p38) [31,32,33]. The importance of the NF-κB pathway in osteoclastogenesis has been widely studied both in vitro and in vivo [34]. NF-κB signaling involves a group of members and a series of activation and/or inhibition steps. Under normal conditions without stimulation, the NF-κB dimer (also known as NF-κB p50/65) exists as a complex with cytoplasmic inhibitory proteins called the nuclear factor of kappa light polypeptide gene enhancer in B-cells inhibitor (IκB) family (IκBα, IκBβ, IκBε, and Bcl-3). With the stimulation of RANKL, IκB kinase (IKK) is activated and stimulates the phosphorylation of IκB proteins at Ser32 and 36 residues, which leads to the polyubiquitination of IκB and subsequent proteasomal degradation. IκB degradation releases and activates NF-κB, which results in a translocation of NF-κB into the nucleus, which increases the transcription potential of genes involved in osteoclast differentiation [33]. 

In this study, ovotransferrin showed inhibition of RANKL-induced activation of NF-κB pathway via attenuating the phosphorylation of IKKα/β (Ser 176/180), IκBα (Ser 32), and NF-κB p65 (Ser 536). Activation of MAPK signaling is another pivotal regulator in the osteoclastogenesis process [4,33]. MAPKs play an essential role in transducing extracellular into intracellular stimuli to respond and contribute to diverse cellular activities [35]. Hence, in the current study, we demonstrated that ovotransferrin markedly prevents RANKL-induced phosphorylation of three major MAPK kinases (ERK, JNK, and p38). These results indicate that the anti-osteoclastogenic effects of ovotransferrin are connected with the suppression of RANKL-induced NF-κB and MAPK signaling pathways.

The phosphorylation of the NF-κB and MAPK signaling pathways is the early stage of osteoclastogenesis. Following the activation of NF-κB and MAPK signaling, the transcription of osteoclast differentiation factors, for example c-Fos and NFATc1, is expressed, acting as the final regulator for osteoclast differentiation [33,36]. Binding of NF-κB and NFATc1 to their promoter in the nucleus controls the expression of several specific osteoclastogenesis genes, including *trap*, *cathepsin K*, and *DC-STAMP* (dendritic cell-specific transmembrane protein) [2,33]. In our study, we observed a decrease in the c-Fos, NFATc1, and cathepsin K expressions in RANKL-stimulated osteoclast precursors when pretreated with ovotransferrin. These results suggest that the reduction of the expression of c-Fos, NFATc1, and their downstream regulators that are crucial for ovotransferrin inhibit osteoclastogenesis.

In addition to inhibiting osteoclast formation, pharmaceutical agents such as bisphosphonates and tamoxifen prevent osteoclastic bone resorption by accelerating cell apoptosis in mature osteoclasts via different mechanisms [8,21]. Bisphosphonates can cause osteoclast apoptosis by inhibiting the activity of farnesyl pyrophosphate synthase, which is a key regulatory enzyme in the mevalonic acid pathway [37]. The activation of this enzyme can interfere with the posttranslational modification of several specific proteins that play central roles in regulating osteoclast survival, such as the guanosine triphosphate-binding proteins Rab, Rac, and Rho [37]. Tamoxifen, acting as estrogen receptor agonist, prevents extreme bone loss by controlling osteoclast life span through the elevation of osteoclast apoptosis mediated through transforming growth factor β (TGF-β) [8]. Thus, in this study, we investigated whether egg white ovotransferrin may affect osteoclast function by inducing apoptosis. We found that ovotransferrin could induce cell apoptosis in mature osteoclasts by affecting the gene expression of *Bcl-2* family members. The mRNA expression of anti-apoptotic proteins *Bcl-2* and *Bcl-xl* in osteoclasts decreased, whereas the pro-apoptotic proteins *Bim* and *Bax* increased in the presence of ovotransferrin.

Generally, the apoptotic process is triggered by two pathways. One is recognized by ligand activation via the death receptor pathway [7]. These death receptors belong to the tumor necrosis factor (TNF) receptor superfamily, including Fas, TNF-related apoptosis-inducing ligand (TRAIL), and TNF-R1 [15]. However, the regulation of the death receptor pathway in osteoclasts shows several differences. For example, TNF promotes osteoclast survival rather than apoptosis in other cells, which might be due to the positive influence of inflammation stimuli on osteoclastogenesis [13,14]. The other pathway is regulated by Bcl-2 family members and their contribution to the mitochondrial release of cytochrome c, which is also known as the mitochondrial pathway [7]. The anti-apoptotic Bcl-2 family members include Bcl-2, Bcl-xL, Bcl-w, Mcl-1, A1, and Boo/Diva, which are mainly responsible for protecting the cell against apoptosis [7]. The pro-apoptotic members include BH3-only pro-apoptotic members (Bim, Bad, and Bid) and other pro-apoptotic proteins (Bak, Bax, and Bod), which act to induce apoptosis [7]. Although the regulatory mechanisms of the Bcl-2 family underlying osteoclasts apoptosis have not been fully identified and characterized, studies have demonstrated the significant relationship between the expression of the Bcl-2 family and osteoclast apoptosis. Among all Bcl-2 family members, anti-apoptotic protein Bcl-xl and pro-apoptotic protein Bim have been most studied in osteoclasts. Over-expression of Bcl-xl in osteoblasts prolongs their life span in vitro and protects them against bisphosphonate-induced cell death, which indicates the adverse effects of Bcl-xl on inducing osteoclast apoptosis and prevention of bone loss [18]. However, other results have been found in vivo. Iwasawa et al. found that the anti-apoptotic protein Bcl-xl positively regulates osteoclast survival, but negatively regulates osteoclast activity by regulating the production of extracellular matrix (ECM) proteins and c-Src kinase activity [38]. Although it is possible that only the inhibitor treatment (ABT-737) used in the study produced the effects, more studies are warranted to validate the effects of Bcl-xl protein, or even the Bcl-2 family proteins, on osteoclasts survival and their ultimate resorbing activity. Compared with Bcl-xl, the reports of the effect of pro-apoptotic protein Bim have been consistent in different studies. The increased number of osteoclasts with an increased life span was observed in Bim-/- mice, which suggests the negative effect of Bim in osteoclast survival [39]. TGF-β1, the key functional modulator of osteoclasts, can cause osteoclast apoptosis by affecting the expression of Bim. The expression of Bim increased in the presence of TGF-β1, which led to the upregulation of activated caspase 9 [40]. In addition to Bcl-xl and Bim, Bcl-2 plays roles in osteoclast apoptosis and pathology of postmenopausal osteoporosis. The mRNA and protein levels of Bcl-2 in osteoclasts are distinctly increased in postmenopausal osteoporosis patients, leading to the inhibition of osteoclast apoptosis and excessive bone loss [41].

## 5. Conclusions

In this study, we established that egg white ovotransferrin could suppress RANKL-mediated osteoclastogenesis and resorption activity via suppression of NF-κB and MAPK activation during the process of osteoclast differentiation along with the induction of cell apoptosis in mature osteoclasts. The balance among osteoclasts, osteoblasts, and osteocytes is an important determinant of bone mass and strength. However, some of the drugs used for osteoporosis prevention by regulating osteoclastogenesis and osteoclast life span affect the survival of osteoblasts and/or osteocytes; therefore, long-term use may be harmful to the skeleton’s integrity. We found that the egg white protein ovotransferrin inhibits osteoclasts differentiation and survival, suggesting its potential for use as a functional food ingredient in bone health management. 

## Figures and Tables

**Figure 1 nutrients-11-02254-f001:**
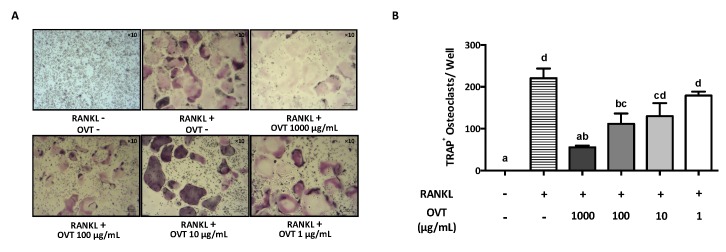
Effect of ovotransferrin on RANKL-induced osteoclastogenesis. (**A**) RAW 264.7 cells were incubated with DMEM containing RANKL (100 ng/mL) and different concentrations of ovotransferrin (OVT) for 6 days before the TRAP staining was performed. All images were captured under 10× magnification. (**B**) TRAP^+^ multinucleated cells with at least three nuclei were identified and counted as mature osteoclasts. The results are expressed as means ± SEM with representative of 6 independent experiments. Means with different letter indicate *p* < 0.05.

**Figure 2 nutrients-11-02254-f002:**
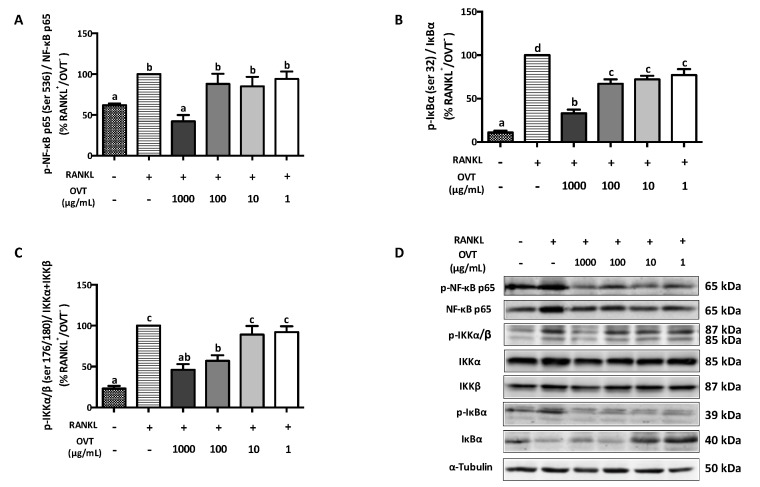
Effect of ovotransferrin on RANKL-induced NF-κB activation. RAW 264.7 cells were pretreated with ovotransferrin (1–1000 μg/mL) for 30 min prior to stimulation with RANKL (100 ng/mL). Then, RAW 264.7 cells were treated with both RANKL and ovotransferrin for 4 h. Whole cell lysates were used for Western blot analysis of (**A**) p-NF-κB p65 (ser 536)/NF-κB p65, (**B**) p-IκB (ser 32)/IκB, and (**C**) p-IKKα/β (ser 176/180)/IKKα+IKKβ. (**D**) All bands are shown. The results are expressed as means ± SEM representative of 4 independent experiments. Means with different letters indicate *p* < 0.05.

**Figure 3 nutrients-11-02254-f003:**
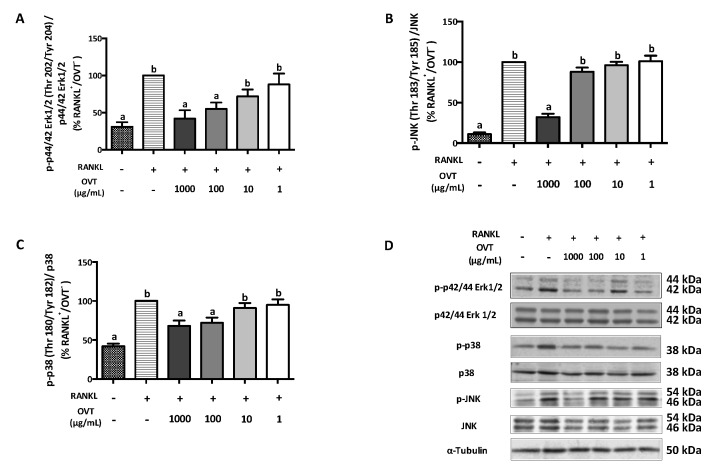
Effect of ovotransferrin on RANKL-induced MAPK activation. RAW 264.7 cells were pretreated with ovotransferrin (1–1000 μg/mL) for 30 min prior to stimulation with RANKL (100 ng/mL). Then, RAW 264.7 cells were treated with both RANKL and ovotransferrin for 4 h. Whole cell lysates were used for Western blot analysis of (**A**) p-p44/42 Erk1/2 (Thr 202/Tyr 204)/p44/42 Erk1/2, (**B**) p-p38 (Thr 180/Tyr182)/p38, and (**C**) p-JNK (Thr 183/Tyr 185)/JNK. (**D**) All bands are shown. The results are expressed as means ± SEM representative of 4 independent experiments. Means with different letters indicate *p* < 0.05.

**Figure 4 nutrients-11-02254-f004:**
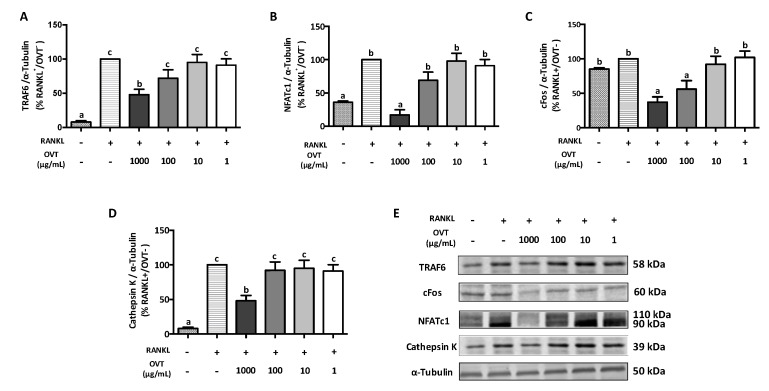
Effect of ovotransferrin on RANKL-induced expression of proteins involved in osteoclastogenesis. RAW 264.7 cells were pretreated with ovotransferrin (1–1000 μg/mL) for 2 h prior to stimulation with RANKL (100 ng/mL). Then, RAW 264.7 cells were co-cultured with RANKL and ovotransferrin for 12 h. Whole cell lysates were used for Western blot analysis of (**A**) TRAF6, (**B**) cFos, (**C**) NFATc1, and (**D**) cathepsin K. (**E**) All bands are shown. The results are expressed as means ± SEM representative of 4 independent experiments. Means with different letters indicate *p* < 0.05.

**Figure 5 nutrients-11-02254-f005:**
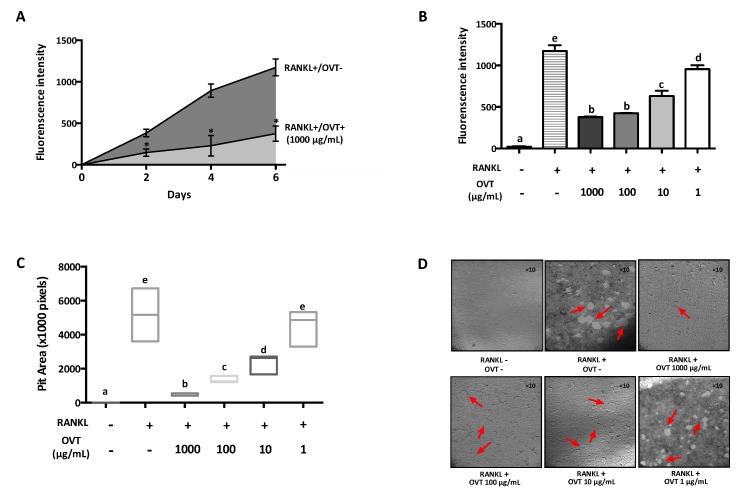
Effect of ovotransferrin on RANKL-induced bone resorption in vitro. RAW 264.7 was cultured onto fluoresceinated calcium phosphate (CaP)-coated plate and stimulated with RANKL (100 ng/mL) for 6 days. After 1 day for adhesion, cells were treated with ovotransferrin (1–1000 μg/mL) and RANKL together for 6 days. (**A**) The fluorescence intensity of resorbed CaP during 6-day treatment. * *p* < 0.05. (**B**) The total resorbed CaP after 6 days. (**C**) The total area of resorption pits after 6 days. (**D**) Image of resorption area. All images were captured under 10× magnification. The results are expressed as means ± SEM representative of 4 independent experiments. Means with different letters indicate *p* < 0.05.

**Figure 6 nutrients-11-02254-f006:**
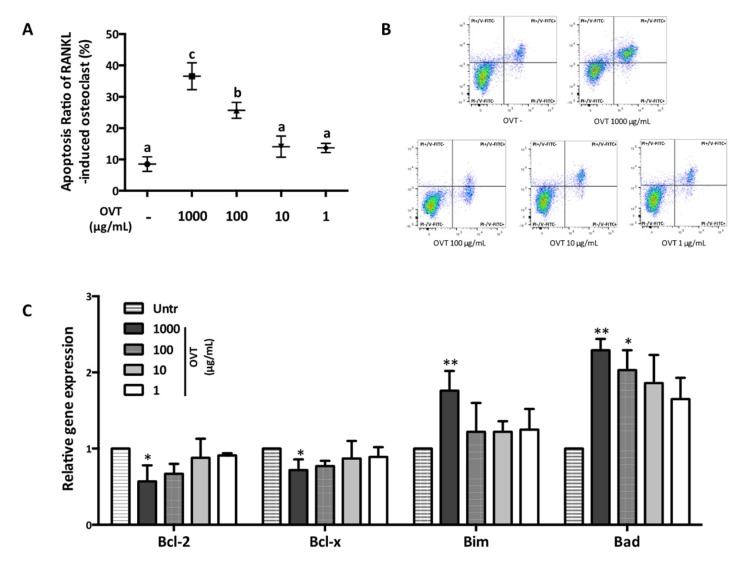
Effect of ovotransferrin on RANKL-induced osteoclasts apoptosis. Osteoclasts were generated by stimulating RAW 264.7 with RANKL (100 ng/mL) for 4–6 days and multinucleated cells were identified by microscopy. Then cells were treated with ovotransferrin (1–1000 μg/mL) for 12 h. (**A**) Osteoclasts apoptosis percentage. Means with different letters indicate *p* < 0.05. (**B**) Osteoclasts apoptosis measured by flow cytometry. The lower left quadrant: PI-/V-FITC-; upper left quadrant: PI+/V-FITC-; lower right quadrant: PI-/V-FITC+; upper right quadrant: PI+/V-FITC+. (**C**) The gene expression of the *Bcl-2* family (*Bcl-2, Bcl-xl, Bim,* and *Bid*). The results are expressed as means ± SEM representative of 4 independent experiments. * *p* < 0.05; ** *p* < 0.01.

**Figure 7 nutrients-11-02254-f007:**
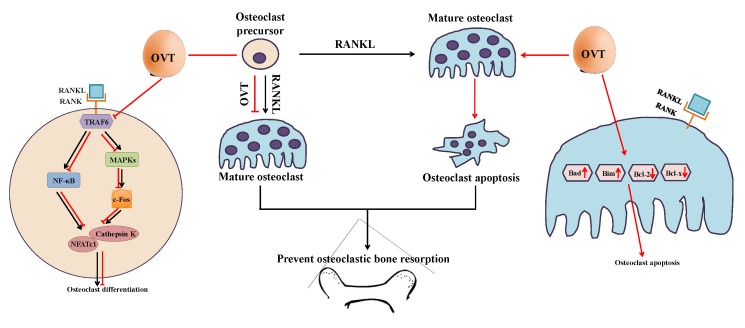
The mechanism of ovotransferrin attenuating RANKL-induced bone resorption. The mature osteoclast differentiates from its precursor and is responsible for bone resorption. OVT inhibits the activation of NF-κB and MAPK pathways during the differentiation, and then down-regulates several proteins (c-Fos, NFATc1, and cathepsin K) involved in osteoclastogenesis. OVT stimulates cell apoptosis in mature osteoclasts via regulating the expression of Bcl-2 family members. Based on this, OVT may have the ability to protect against bone loss.

**Table 1 nutrients-11-02254-t001:** Sequences of primers used in real-time PCR.

Gene	Primer Sequences (5’-3’)	
*Bcl-2*	Forward	TGAACCGGCATCTGCACAC
	Reverse	CGTCTTCAGAGACAGCCAGGAG
*Bcl-x*	Forward	GCTGGGACACTTTTGTGGAT
	Reverse	TGTCTGGTCACTTCCGACTG
*Bax*	Forward	ACCAGCTCTGAACAGATCATG
	Reverse	ACTTTAGTGCACAGGGCCTTG
*Bim*	Forward	CTTCCATACGACAGTCTC
	Reverse	AACCATTTGAGGGTGGTCTTC
*β-actin*	Forward	AGATGTGGATCAGCAAGCAG
	Reverse	GCGCAAGTTAGGTTTTGTCA

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
