# Peer review of "Egg White Ovotransferrin Attenuates RANKL-Induced Osteoclastogenesis and Bone Resorption"

_nutrients, 2019, doi:10.3390/nu11092254_

Round 1
Reviewer 1 Report
I was pleased to review the manuscript entitled “Egg White Ovotransferrin Attenuates RANKL-induced Osteoclastogenesis and Bone Resorption”. The paper is an in vitro study in which the biological effects of ovotransferrin on murine monocyte cell line were investigated by means of specific assays for osteoclast differentiation and function, western blot and RT-PCR. Overall, the manuscript is original and novel, methods are sound and quite clearly described.
My major concern is about the reporting of results in light of the performed statistical analysis. Troughs the results, the authors reported in a generic way that ovotransferrin “inhibits osteoclast formation or RANKL-induced activation of NF-kB and MAPK pathways or resorption activity” and widely used the term “in a dose dependent manner”. However, the statistical analysis did not show this. As examples, TRAP results highlighted that ovotransferrin at concentrations of 1000 and 100 µg/ml significantly decreased the number of TRAP stained osteoclasts, but not the concentrations of 10 and 1 µg/ml that did not differ from control. In NF-kB, p-JNK, TRAF6, NFATc1, Cathepsin K and others, only ovotransferrin at 1000 µg/ml exerted significant different results than control but not the other tested concentrations. So far, results (and thus discussion) should be re-written in a critical and objective fashion in light of the obtained experimental results and performed statistical analysis.
Minor suggestions/concerns are as follows:
- Please move the last 2 sentences of the Introduction to the Discussion section since it reports obtained results.
- In the cell apoptosis assay, cells were collected for flow cytometry. How were they collected?
-Western blot analysis: please provide specifications of antibodies used and their concentrations. Moreover, in the method, 1000 µg/ml of lactoferrin were reported to be used, but this group is not reported in the results and figure. Please add results and specify why lactoferrin was used.
-The sentence at page 3 lines 124-126 “After incubation…as described previously.” needs the reference.
-For TRAP and Western blot assays, the same cell seeding density is reported (1 x 104 cell/well) but 24 and 48 multiwell plates were used. Is it correct?Moreover, for Western blot cells were incubated in ∝-MEM whereas troughs the manuscript DMEM was used. Why these changes of the culture conditions? How were results comparable?
-In the statistical analysis ANOVA with Dunnett’s post-hoc test was used “for comparison to control”. Did the authors mean to cells in presence of only RANKL, as control? However, in figures the significance is not only to control but also among the different concentrations of ovotransferrin used.
-Figures 1 and 5. Microscopic images lack of the bar or it is too small to be readable. Moreover, in the legend please provide magnifications.
-Results presented by the authors seem discordant with the paper of Lee JH et al. (Korean J Food Sci Anim Resour. 2018 Dec;38(6):1226-1236) that reported the role of ovotransferrin in increasing pro-inflammatory cytokine secretion and MAPK signalling. Can the authors discuss their results in a comparative manner with this and other relevant papers.
Reviewer 2 Report
Review 08/21/19
Title: Egg White Ovotransferrin Attenuates RANKL-induced
Osteoclastogenesis and Bone Resorption
Journal: Nutrients
Summary:
The authors report on a study in which they evaluate the role of ovotransferrin on differentation and apoptosis in bone cells. The authors provide evidence that suggests that ovotransferrin inhibits some functions of osteoclasts in an in vitro setting, including differentiation, calcium resorption and induced apoptosis of mature osteoclasts.
Introduction:
Typo in line 24 - "...formation that regulated by" -> should be [is regulated by...]
It would be nice to include a summary figure of the osteoclast molecular pathways described in the 2nd paragraph.
Methods:
In an effort for science to be more reproducible, you sould report unique identifiers for key reagents used in this study:
- report the catalog number for the RANKL (and the gene ID from Entrez Gene)
- report the catalog number for the cell line
- report the catalog number for the TRAP staining kit and apoptosis detection kit
Where was the ovotransferrin obtained from? Report the manufacturer and catalog number.
The Western blot assay states the methods were described previously, but there is no reference.
Report the following information on your westerns at a minimum:
- What antibodies were used?
- What antibody was used for the loading control?
- report the species of origin
- report the antibody name, manufacturer, catalog number and lot number and dilution of each antibody.
- What were the antibodies diluted into?
- How were the blots analyzed for the loading control - were the antibodies stripped and reprobed with anti-alpha-tubulin?
Results:
There is not any report on the validity of the antibodies. How can you ensure the antibody is binding to the intended target? Especially the phospho antibody, this should be validated and reported.
Figure 2 and 3, does the loading control correspond to each blot in the image? Or was there a separate loading control for each blot?
Molecular weight markers should be included in the blots.
What do the letters a,b,c, and d indicate in Figure 2?
In Figure 3, there are multiple bands for JNK and pJNK - which band corresponds to the protein of interest? How did you validate this?
There is a lot of information available on proper repporting for Western date, for example see:
http://www.jbc.org/content/290/50/29692.full
Figure 6 - You need to add figure legends on the scatter plots.
It is quadrant, not quadrand.
Discussion:
You mention that egg whites could be a nutritional therapeutic for bone health, but did not mention how much ovotransferrin is present in typical eggs. Please discuss the feasibility of adding white eggs to your diet as a nutritional intervention - how many eggs would you need to consume to get this therapeutic dose?
Overall:
This paper needs to be proofread and edited for grammar.
To improve the reliability and transparency of these research results, the authors should share their raw data, and unaltered images from the western blot images. Many of the blots show multiple bands, and validation studies should be reported to verify the reliability of their antibodies.
Round 2
Reviewer 2 Report
Thank you to the authors for their revisions of the manuscript. I appreciate that they addressed many of my concerns, but I still do have some issues, discussed below.
Section 2.1
The reported Gene ID is not the ID, they reported the label. The Gene ID would be 21943, I believe (see https://www.ncbi.nlm.nih.gov/gene/21943).
This sentence should be revised to:
Recombinant mouse soluble RANKL (Gene ID: 21943) was purchased from R&D Systems (Catalog number: (462-TEC-010), Oakville, ON, Canada).
The catalog numbers should be reported after the manufacturer's name, and should be prefixed with catalog number, so it is clear what these numbers are referring to.
Section 2.6
You must report the dilutions for each antibody. Sometimes the manufacturer's suggest a range of dilutions, or the dilutions will change. In order for your experiment to be reproducible, readers must know the exact conditions that were used. You could create a table that lists the antibodies and dilutions used.
Regarding the antibody validation: In my opinion, antibodies must be validated in the lab and in the particular system being studied. Even though the antibodies were validated by the manufacturer and in other published studies does not ensure the antibodies are functioning in the same way under your experimental conditions.
Without proper validation of the antibodies in these experimental conditions, I cannot trust the evidence provided in this paper.
Antibodies are notoriously problematic and can bind to off target proteins. It is necessary to validate all antibodies in house for each experiment, one cannot just rely on the manufacturer's or previous group's validation.
The authors stated the raw data and images are available and they would make them publicly available, but it does not appear they have done so.
Without in-house validation of the antibodies used and the unaltered Western blot images, I cannot put any confidence into the validity of these results.
